



# High emission rates and strong temperature response make boreal wetlands a large source of terpenes

Lejish Vettikkat[1], Pasi Miettinen[1], Angela Buchholz[1], Pekka Rantala[2], Hao Yu[3], Simon Schallhart[4], Roger Seco[5], Elisa Männistö[6], Eeva-Stiina Tuittila[6], Alex B. Guenther[7] and Siegfried Schobesberger[1]

[1]Department of Applied Physics, University of Eastern Finland, Kuopio, P.O. Box 1627, FI-70211, Finland
[2]Institute for Atmospheric and Earth System Research, University of Helsinki, Helsinki, 00014, Finland
[3]Department of Environmental and Biological Sciences, University of Eastern Finland, Kuopio, P.O. Box 1627, FI-70211, Finland
[4]Finnish Meteorological Institute, P.O.Box 503, FI-00101 Helsinki, Finland
[5]Institute of Environmental Assessment and Water Research (IDAEA-CSIC), 08034 Barcelona, Catalonia, Spain
[6]Peatland and soil ecology research group, School of Forest Sciences, University of Eastern Finland, P.O. Box 111, 80101 Joensuu, Finland
[7]Department of Earth System Science, University of California, Irvine, Irvine, California, USA

*Correspondence to*: Lejish Vettikkat (lejivett@uef.fi) and Siegfried Schobesberger (siegfried.schobesberger.uef.fi)

**Abstract.** Terpenes, a class of hydrocarbons including isoprene ($C_5H_8$), monoterpenes (MTs; $C_{10}H_{16}$), sesquiterpenes (SQTs; $C_{15}H_{24}$), and diterpenes (DTs; $C_{20}H_{32}$), are highly reactive to atmospheric oxidants and can form highly oxidized organic molecules (HOMs), leading to new particle formation (NPF), and secondary pollutants like tropospheric ozone and secondary organic aerosols (SOA). Wetlands are primarily found in the boreal and tundra regions of the Northern hemisphere and are well-known for their high methane emissions. However, their VOC emissions were investigated by

relatively few studies, showing them to be high isoprene emitters. Terpene emissions have an exponential temperature dependence, while boreal wetlands are experiencing above two times the average global warming, and recent enclosure studies show a substantial increase in terpene emissions.

In this study, we present ecosystem-scale eddy covariance (EC) fluxes of terpenes (up to DTs) from Siikaneva, a boreal fen in southern Finland, from the start to the peak of the growing season of 2021 (19 May 2021 to 28 June 2021). These

are the first EC fluxes reported using the novel state-of-the-art Vocus- proton transfer reaction mass spectrometer (Vocus-PTR) and the first-ever fluxes reported for DTs from a wetland. Isoprene was the dominant terpene emitted by the wetland, followed by MTs, SQTs, and DTs. All terpenes exhibited distinct diurnal patterns with maxima around noon and a strong exponential dependence on temperature. The $Q_{10}$ values, the factor by which terpene emissions increases for every 10 °C rise in temperature, were up to 5 times higher than those used in most BVOC models. During the campaign, the air

temperature peaked above 31 °C on 21-22 June 2021, which is abnormally high for boreal environments, and the maximum flux for all terpenes coincided with this period. We observed that terpene emissions were elevated after this abnormally "high-temperature stress period," indicating that past temperatures alter emissions significantly.

The standardized emission factor (EF) of the fen for isoprene ($EF_{iso}$) was $11.1 \pm 0.3$ nmol m$^{-2}$ s$^{-1}$, which is at least two times higher than in previous studies and as high as the emission factors typical for broadleaf and other forests in the

lower latitudes. We observed $EF_{MT}$ of $2.4 \pm 0.1$ nmol m$^{-2}$ s$^{-1}$, $EF_{SQT}$ of $1.3 \pm 0.03$ nmol m$^{-2}$ s$^{-1}$, higher than typical for needle leaf and broadleaf tree functional types, and $EF_{DT}$ of $0.011 \pm 0.001$ nmol m$^{-2}$ s$^{-1}$. We also compared the landscape average emissions to the Model of Emissions of Gases and Aerosols from Nature (MEGAN) v2.1, specifically using EF for the "C3 Arctic grass" plant functional type, and found that the emissions were underestimated by over 9 times for isoprene, over 300 times for MTs, and 800 times for SQTs. Our results show that due to very high EFs and high sensitivity





to increasing temperatures, these high latitude ecosystems can be a large source of terpenes to the atmosphere, and anthropogenic global warming could induce much higher BVOC emissions from wetlands in the future.

## 1 Introduction

The emissions of biogenic volatile organic compounds (BVOCs) from terrestrial ecosystems to the atmosphere are estimated to be around 1 Pg ($10^{15}$ g) per year (Guenther et al., 2012). Among these BVOCs, terpenes are a class of

hydrocarbons with the formula $(C_5H_8)_n$, namely, isoprene ($C_5H_8$), monoterpenes (MTs; $C_{10}H_{16}$), sesquiterpenes (SQTs; $C_{15}H_{24}$), diterpenes (DTs; $C_{20}H_{32}$), and even larger, more complex compounds. Terpenes account for 70 % of total global VOC emissions (Guenther et al., 1995; Guenther et al., 2012). Terpenes, except isoprene, have a huge variety of structures. They contain one or more double bonds making them highly reactive. Once emitted by plants, terpenes affect the oxidative capacity of the atmosphere by reacting with oxidants such as hydroxyl radicals (OH), ozone ($O_3$), or nitrate radicals ($NO_3$),

forming less volatile oxygenated VOCs (OVOCs) (Atkinson and Arey, 2003) that may be able to condense onto existing aerosol particles and contribute to secondary organic aerosol (SOA) mass (Kroll and Seinfeld, 2008; Hallquist et al., 2009). Highly oxidized organic molecules (HOMs) formed by autoxidation of terpenes can also lead to new particle formation (NPF) (Ehn et al., 2014; Kirkby et al., 2016; Bianchi et al., 2019). SOA particles can act as cloud condensation nuclei (CCN) that affect Earth's climate (Kulmala et al., 2013; IPCC 2021). In polluted environments rich in nitric oxide

(NO), organic peroxyl ($RO_2$) radicals formed by the oxidation of terpenes can fuel tropospheric ozone formation (Jacob, 1999). Terpenes also play a key ecological role in protecting plants from both biotic and abiotic stresses such as high temperature (Sharkey and Singsaas, 1995; Loreto et al., 1998), intense light (Vickers et al., 2009), and herbivory (Kappers et al., 2011).

Temperature and photosynthetically active radiation (PAR) are the main drivers of terpene emissions, and the emissions

have an exponential dependence on temperature when PAR is saturated (Niinemets et al., 2004). Hence, emissions of terpenes are modeled using algorithms based on the leaf-level response of emissions to the variations in temperature and PAR (Guenther et al., 2012; Monson et al., 2012). Common BVOC models like the Model of Emissions of Gases and Aerosols from Nature (MEGAN) estimate emissions as the product of emission activity factors and standardized emission factors representing important plant functional types (Guenther et al., 2012). Wetlands cover about 2% of the global land

surface area, and most of these wetlands are found in the boreal and tundra regions of the Northern hemisphere (Archibold, 1995). These northern latitudes are experiencing above two times the average global warming, and it is certain (with high confidence) that northern latitudes, especially the Arctic, will continue to experience this warming (IPCC 2021; Post et al., 2019). Due to increased warming, these ecosystems will respond with increased terpene emissions. As an indirect effect, warming can also change the vegetation composition in these ecosystems (Valolahti et al., 2015). Furthermore, the

impact of terpenes is generally more critical in high latitudes because of low anthropogenic VOC emissions (Paasonen et al., 2013).

There is an extensive assortment of studies of greenhouse gas emissions from high latitude wetlands (Aurela et al., 2007; Rinne et al., 2007; Rinne et al., 2018). They are well known as a sink of carbon dioxide and the largest natural methane source to the atmosphere. In contrast, their VOC emissions were investigated by relatively few studies, which have shown

wetlands to be high isoprene emitters (Janson and De Serves, 1998; Haapanala et al., 2006; Hellén et al., 2006; Ekberg et al., 2009). Most of these studies were conducted using chamber/enclosure measurements and have not investigated the emission of other terpenes. Recent enclosure studies have shown a substantial increase in isoprene emissions in Arctic tundra heath and subarctic wetlands in response to warming (Kramshøj et al., 2016; Lindwall et al., 2016). Disadvantages





of such chamber measurements are the rise in temperature and humidity inside the enclosure (Ortega and Helmig, 2008),
the acclimation and stress issues for the vegetation inside the enclosure (Niinemets et al., 2011), and the potential loss of
less volatile or highly reactive vapors to the enclosure walls. Eddy covariance (EC) is an ecosystem-scale flux
measurement technique widely used to measure greenhouse gas fluxes (Aubinet et al., 2012). This micrometeorological
technique overcomes most disadvantages of chamber measurements and directly assesses ecosystem-level fluxes
compared to up-scaling fluxes measured from small enclosures. Very few ecosystem-scale BVOC flux measurement
studies have been conducted in wetland ecosystems (Seco et al., 2020; Holst et al., 2010). They found isoprene emissions
to have a steeper response to temperature than that used in standard BVOC emission models (Guenther et al., 1993;
Monson et al., 2012), indicating a particularly high-temperature sensitivity of arctic vegetation due to their acclimatization
to colder temperatures. However, none of those studies had the analytical capability to measure fluxes of terpenes larger
than MTs.

Isoprene is the most emitted BVOC globally, and its oxidation chemistry is well studied and is shown to contribute to
SOA mass (Henze and Seinfeld, 2006). MTs and SQTs oxidation products are well known to partition into the aerosol
particle phase. Highly reactive SQT could significantly contribute to SOA mass despite generally lower emissions than
MTs (Barreira et al., 2021; Hellén et al., 2018). Measurements of SQT and DT are particularly challenging because
sampling loss issues are aggravated by their lower volatility and generally higher reactivity than the smaller terpenes.
However, a recent chamber study measured SQT from a wetland ecosystem and observed their emissions exceeding the
MTs emissions (Hellén et al., 2020). Since diterpenes have very low volatility, they were not thought to be emitted by
terrestrial vegetation (Guenther, 2002). Matsunaga et al. (2012) were the first to report DT emissions from vegetation in
enclosure measurements. Recent developments in mass spectrometric techniques have enabled measurements of DTs in
ambient air (Li et al., 2020). Only one study has reported ecosystem-scale DT emission flux (Fischer et al., 2021) for a
boreal forest, but they could not characterize the emissions in terms of temperature and PAR. Due to their high reactivity,
DTs could play an important role in atmospheric chemistry. The high molecular weight of their oxidation products makes
them relevant for SOA mass formation and NPF. Clearly, more emission studies on the larger terpene families are
warranted, given their chemical diversities and important roles in NPF and SOA formation.

We focus here on the characterization of isoprene, MTs, SQTs, and the rarely reported DTs. Together these four groups
are collectively referred to as terpenes throughout this paper. We report ecosystem-scale fluxes of terpenes measured by
the eddy covariance (EC) technique from a boreal fen dominated by sedges using a recently developed Vocus proton
transfer reaction mass spectrometer (Vocus-PTR) (Krechmer et al., 2018), from the start to the peak of the growing season
in spring and early summer. This instrument has very high sensitivity compared to classical PTR-MS; hence we could
measure the first DT emission fluxes from this ecosystem. The terpene emissions are parametrized by temperature, PAR,
and leaf area index (LAI). We also calculate the temperature dependence of the emissions and the standardized emission
factor (EF) of the terpenes for this ecosystem. Finally, we compare the measured terpene emissions with the MEGANv2.1
emission model and their temperature dependence.

## 2 Methods

### 2.1 Site description

The eddy covariance measurements were conducted at the Siikaneva 1 site of the Station for Measuring Ecosystem-
Atmosphere Relations II (SMEAR II). Siikaneva is an oligotrophic fen located 5 km west of the Hyytiälä Forestry Field





Station in southern Finland (61°49'60" N 24°11'32"E; 162 m a.s.l.). The vegetation in the ground layer is dominated by peat mosses (*Sphagnum balticum*, *S. papillosum, S. magellanicum*, and *S. majus*), while the dominating vascular plant species are sedges *Carex rostrata*, *C. limosa*, *C. lasiocarpa*, and *Eriophorum vaginatum*. The microtopography consists

of different plant community types that vary from wet hollows to intermediated sedge-dominated lawns to dry hummocks dominated by dwarf shrubs (*Andromeda polifolia*, *Betula nan*a, *Rubus chamaemorus,* and *Vaccinium oxycoccus*) for 200 m to 400 m in all directions, and the fen is surrounded by Scots pine forest. A comprehensive vegetation inventory of the site was conducted during the peak growing season in 2017. The annual mean temperature from 1981-2010 was 4.2 °C, and the annual total precipitation was 711 mm (Pirinen et al., 2012). Siikaneva has been a well-established site for eddy

covariance measurements of greenhouse gases since 2005 (Aurela et al., 2007; Rinne et al., 2007) and is also part of the integrated carbon observation system (ICOS) network since 2017 (Rinne et al., 2018).

The EC measurements took place in the growing season from late spring to early summer (from 19 May to 28 June 2021), 2.4 m above the fen. Figure S1 shows a satellite view of the sampling site, overlaying with the average footprint calculated using a two-dimensional model (Kljun et al., 2015). Over 90% of the campaign-average flux footprint is within the

wetland area and ~150 m from the sampling site. Therefore, the flux results have a negligible contribution from the surrounding Scots pine forest. Meteorological parameters such as air temperature, photosynthetically active radiation (PAR), soil moisture, and humidity were obtained from the SmartSMEAR website (https://smear.avaa.csc.fi/; last access: 01 February 2022; (Junninen et al., 2009). Air temperature and relative humidity (RH) were measured with an HC2 sensor (Rotronic AG, Switzerland) at a 2-meter height in Siikaneva, and PAR was measured by a Li-190SZ quantum sensor (LI-

COR, Inc., USA). Leaf area index (LAI) was modeled and calculated for the 90 % contribution of flux footprint area during the measurement campaign separately for shrubs (LAIshrub), other vascular plants (LAIother), and all vascular plants (LAItotal). LAI was first calculated for each plant community type of the site (see Table S1). Leaf count and average leaf size for each vascular plant species were conducted every third week from permanent ICOS study plots. The area-based LAI of each vascular plant species per plot was then calculated by multiplying leaf count by the average leaf

size. Finally, area-based LAI was weighted with their portion in the flux footprint (Table S1) to obtain LAIshrub, LAIother, and LAItotal of the footprint area. Table S1 also shows the mean moss cover as % in the footprint.

**2.2 Eddy covariance measurement setup**

The Vocus-PTR was placed on a wooden platform at the Siikaneva 1 site. We adapted the inlet design from Fischer et al. (2021), who used a similar setup and essentially the same sampling strategy for EC flux measurements. A high-flow main

inlet (galvanized steel, 20 cm I.D.) with 5000 lpm suction flow (CK-200A blower, Onnline Oy, Sweden) sampled air from 2.4 m above the fen. A sonic anemometer (METEK USA-1) for measuring the vertical and horizontal components of the wind vector was mounted 0.4 m above the main inlet to minimize the disturbance from the high inlet flow. A horizontal core sub-sampling (Teflon tubing, 10mm I. D.) of 5 lpm perpendicular to the main inlet was drawn from within the entrance cone of the main flow to the Vocus-PTR, which minimized sample interactions with the main inlet walls and

also substantially reduced subsequent wall losses of less volatile compounds such as SQT, DTs, and oxygenated VOCs, as only 100 sccm of the sample flow would enter the instrument reaction region. Figure 1 shows the setup used here, including a schematic of the flows involved in sampling, calibrating, and zeroing.

The details about the Vocus-PTR calibrations, the data pre-processing, and the EC flux calculations with the innFLUX code package (Striednig et al., 2020) are provided in SI sections S1.1 to S1.5.






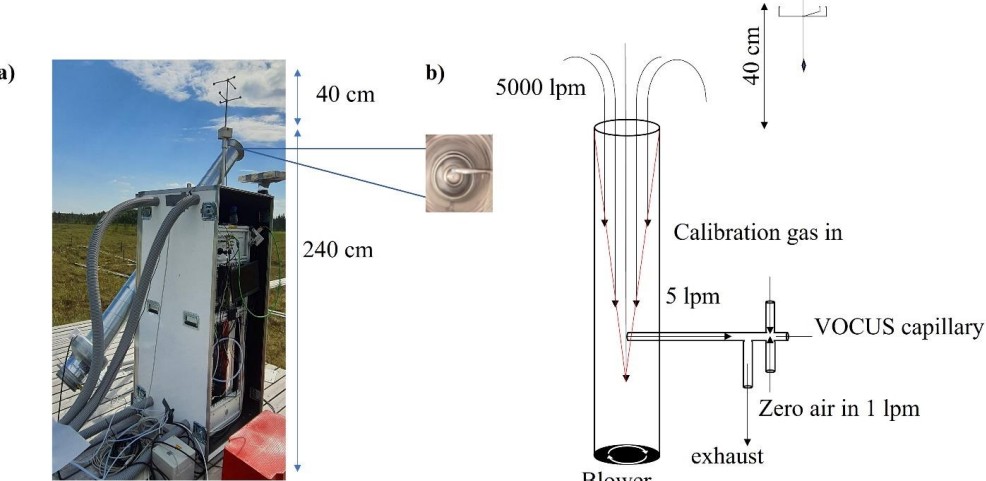

**Figure 1.** a) Eddy covariance flux measurement setup with high-flow inlet with the core sampling inlet visible when looking straight into the main inlet (inset), and b) schematic of the flows involved in sampling, calibrating and zeroing.

### 2.3 Vocus proton transfer reaction time of flight mass spectrometer (Vocus-PTR)

BVOC measurements at a high time resolution of 10 Hz were performed using a Vocus-PTR, described in detail in Krechmer et al. (2018). This novel instrument uses PTR as an ionization technique but differs from conventional PTR in many ways. The key differences are described briefly.

Firstly, a low-pressure (1.5 mbar) focusing ion molecular reactor (FIMR) uses a radiofrequency (RF) field to focus the ions on the central axis. This drastically improves the detection efficiency of product ions. Furthermore, the sensitivity of Vocus-PTR is independent of ambient humidity due to the high water mixing ratio in the FIMR (up to 15% v/v). Of the 5 lpm sampled from the core of the main inlet, only 100 sccm are subsampled (again from the core of the flow) using a capillary inlet, while the rest is directed to exhaust (Fig. 2). This further reduces wall losses and improves the sensitivity of less volatile compounds. The ToF mass analyzer had a mass resolving power of >10000 m/dm throughout the campaign, enabling high-resolution peak identification and typically unambiguous attributions of elemental formulas. The Vocus-PTR also has an inbuilt dynamic dilution setup for automated calibration. We operated the FIMR at a pressure of 1.5 mbar, a temperature of 50 °C, and an electric field of 40 V cm$^{-1}$, resulting in field strength (E/N) of 119 Townsend ($10^{-17}$ V cm$^2$) (Krechmer et al., 2018).

### 2.4 Gas chromatography-mass spectrometry (GC-MS)

We compared the average BVOC concentration from Vocus-PTR measurements with offline GC-MS samples collected on 28 June 2021, the last day of the campaign. Ambient air was sampled through three cleaned Stainless-steel tubes (Markes Intl., UK) filled with adsorbents (Tenax TA and Carbograph 5, 100 mg of each, mesh 60/80, Supelco, Bellefonte, PA, USA) at different locations: close to the wetland surface (~5 cm height), on top of Vocus-PTR box (2 m height), and near the surface of the wooden platform (~10 cm height). Specifically, using handheld pumps, 160 sccm of flow was directly drawn through the adsorbent tubes for 30 minutes without pre-treatment, such as an ozone trap. After collection,





the tubes were analyzed by GC-MS (GCMS-QP2020, SHIMADZU, Kyoto, Japan) with a split mode (1:20). Figure S5 compares the terpene mixing ratios measured by the GC-MS at different locations and the terpene mixing ratios measured using the Vocus-PTR during the same period.

We observed a steep concentration gradient in isoprene concentration with vertical height, indicating very high nearby
emissions. Near the wooden platform and close to the wetland, the isoprene mixing ratio was about 4-5 ppbv (parts per billion by volume) and was reduced to half on top of the Vocus (2 m above the wetland). However, the mixing ratios for isoprene obtained from the adsorbent tube on top of the Vocus (40 cm below the high flow inlet) were comparable to that of the Vocus-PTR (1.7 vs. 1.4 ppbv).

We detected α-pinene, 3-carene, β-pinene, camphene, and limonene in the GC-MS analysis. Table S2 shows the different
monoterpenes and their amount in the three adsorbent tubes (in ng). α-pinene was the most abundant monoterpene, followed by 3-carene. We compared the sum of speciated MTs measured by GC-MS with the sum of all MTs measured by Vocus-PTR (from the parent ion signal ($C_{10}H_{17}^{+}$) using the α-pinene sensitivity). The GC-MS obtained a mixing ratio of 300 pptv, whereas Vocus measured 580 pptv for MTs. The GC-MS MTs mixing ratio may be underestimated due to not detecting monoterpenes other than in Table S2 and oxidation of sampled MT, e.g., by $O_3$ (Helin et al., 2020). In
addition, the Vocus-PTR might overestimate the mixing ratios due to the possibility of other terpenes fragmenting to $C_{10}H_{17}^{+}$. The GC-MS could not reliably detect any SQT, although we detected 100 pptv of sesquiterpenes in the VOCUS-PTR. Since the mixing ratios of isoprene near the inlet agreed between both methods within the different analytical capabilities of the two instruments, we believe the difference in mixing ratios of the larger terpenes could be due to the unintended loss of these compounds with low volatility and high reactivity.

**2.5 Emission modeling**

Global BVOC emission models like the Model for Emissions of Gases and Aerosols from Nature (MEGAN) parametrize the emission rate of a terpene, i, ($E_i$), by the emission factor at standardized conditions ($EF_i$ at T=30 °C, PAR=1000 µmol $m^{-2}$ $s^{-1}$) and is driven by light and temperature. In a pre-MEGAN version (hereafter referred to as "G93"), the isoprene emission is driven by instantaneous leaf temperature and PAR, and MT emissions are driven only by leaf temperature.
The G93 model was developed for individual leaves but can be used for a whole canopy by treating vegetation as one big leaf and so does not consider LAI (Guenther et al., 1993). In the canopy-scale MEGANv2.1 model, the emissions of isoprene, MT, and SQT are driven by instantaneous and past one-day and 10-day leaf temperature and PAR. However, DTs are not included in the model. MEGANv2.1 takes LAI into account, using a canopy environmental model, and the emissions of MT and SQT have both light-dependent and light-independent fractions. The standard conditions for the
MEGANv2.1 are LAI of 5 and average canopy environmental conditions of the past 24 to 240 h leaf temperature of 24 °C and PAR of 200 µmol $m^{-2}$ $s^{-1}$ for sunlit leaves. We use an Excel-based version of MEGANv2.1 for a single location to model the terpene emissions at our Siikaneva site, hereafter referred to as "MEGANv2.1". The parameters and EF values for Arctic C3 grass were used to compare our measured emissions to the model predictions. We did not consider the variations in soil moisture and $CO_2$ for these calculations. The details about the MEGANv2.1 framework are given in
Guenther et al. (2012).

To calculate the EF from our dataset, we used G93 for isoprene and MT and a simplified version of MEGANv2.1, hereafter referred to as G2012, without considering any canopy environment model and assuming all leaves to be sunlit for isoprene, MT, and SQT. Briefly, the emission rate of a terpene, i, ($E_i$), is characterized by its $EF_i$, LAI, and the light ($C_L$) and temperature ($C_T$) activity factors, as shown in Eq 1.



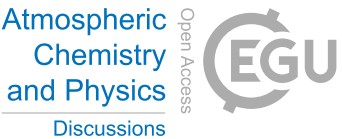

$$E_i = EF_i \cdot LAI \cdot C_L \cdot C_T \quad (1)$$

The explicit parametrizations for the light and temperature activity factors of G93 and G2012 are given below.

$$C_L(G93) = (1 - LDF) + LDF \cdot \frac{0.0027 \cdot 1.066 \cdot PAR}{\sqrt{1 + 0.0027^2 \cdot PAR^2}} \quad (2)$$

$$C_L(G2012) = (1 - LDF) + LDF \cdot \frac{C_P \cdot \alpha \cdot PAR}{\sqrt{1 + \alpha^2 \cdot PAR^2}} \quad (3)$$

$$C_T(G93) = (1 - LDF) \cdot exp^{\beta \cdot (T-303)} + LDF \cdot \frac{exp^{\frac{CT1 \cdot (T-303K)}{R \cdot 303K \cdot T}}}{1 + exp^{\frac{CT2 \cdot (T-313K)}{R \cdot 303K \cdot T}}} \quad (4)$$

$$C_T(G2012) = (1 - LDF) \cdot exp^{\beta \cdot (T-303)} + LDF \cdot E_{opt} \frac{CT2 \cdot exp^{CT1 \cdot X}}{CT2 - CT1 \cdot (1 - exp^{230 \cdot X})} \quad (5)$$

where R is the gas constant, CT2 is 231 kJ mol[-1], and LDF is the light-dependent fraction (between 0 and 1). For G93, the LDF is set to 1 for isoprene and 0 for MT and omits SQT. LDF for G2012 is set to 1 for isoprene, 0.525 for MT, and 0.6 for SQT. Since LDF is 0 for MT in G93, $C_T$ is an exponential function of T, and $C_L$ is 1. For isoprene in the G93 algorithm and all compound groups in the G2012 algorithm, the $C_L$ is a sigmoidal function of PAR, and $C_T$ is an

exponential response function of temperature with a maximum at around 35-40 °C.

The light-dependent terms of $C_L$ and $C_T$ in G93 are based on the instantaneous leaf temperature (T) and PAR, whereas the G2012 version of the $C_L$ and $C_T$ emission activity algorithm includes the instantaneous conditions along with their one- and 10-day histories as the average values for those periods (P24, P240, T24, and T240). The parameters describing the impact of past temperature and light conditions in the G2012 algorithm are calculated with the following equations:

$$\alpha = 0.004 - 0.0005 \, \ln(P240) \quad (6)$$

$$C_P = 0.0468 \cdot exp^{0.0005 \cdot (P24 - 200)} \cdot P240^{0.6} \quad (7)$$

$$E_{opt} = Ceo \cdot exp^{0.05 \cdot (T24 - 297K)} \cdot exp^{0.05 \cdot (T240 - 297K)} \quad (8)$$

$$x = \frac{\frac{1}{Topt} - \frac{1}{T}}{R} \quad (9)$$

$$Topt = 313 + 0.6 \cdot (T240 - 297K) \quad (10)$$

β, CT1, and Ceo are empirically determined coefficients for each emission chemical class. β is 0.13 for isoprene, 0.1 for MTs, and 0.17 for SQTs. CT1 is 95 kJ mol[-1] for isoprene, 80 kJ mol[-1] for MTs, and 130 kJ mol[-1] for SQTs. Ceo is 2 for isoprene and 1.83 for MTs, and 2.37 for SQTs. Eq 1, including and excluding LAI, was used to determine the EF of different terpenes from the wetland.

## 3 Results and discussion

### 3.1 Meteorology and overview

We performed the eddy covariance flux measurements in Siikaneva during the growing season, from 19 May to 28 June 2021. The minimum air temperature during the campaign was -1.3 °C, measured in the early morning of 25 May 2021, and the maximum air temperature was 31.7 °C, observed in the afternoon on 22 June 2021. The average air temperature during the campaign was 15.2 ± 6.5 °C. The PAR flux varied from 0-1888 μmol m[-2] s[-1] and the campaign average was

463 ± 493 μmol m[-2] s[-1]. The soil moisture was very high and only varied between 0.76-0.87 m³ of water per m³ of soil. The water table level varied from -12 to 1.3 cm. Thunderstorms accompanied occasional rainfall, and the measurements were interrupted due to power failures on three occasions. There was another interruption from 4th June afternoon to 7th June midnight due to a blockage of the Vocus-PTR capillary. Due to these interruptions, we obtained Vocus-PTR data



for 60% of the 41 days of the campaign. We calculated the leaf temperature for the fen using a canopy environment model (Goudriaan and Van Laar, 1994; Leuning, 1997) using grass as the vegetation type with five canopy layers. The minimum leaf temperature during the campaign was -3.1 C, calculated for the early morning of 25 May 2021, and the maximum leaf temperature was 31.9 °C, calculated for the afternoon of 22 June 2021. The average leaf temperature during the campaign was 15.3 ± 6.8 °C. The estimated leaf temperature is typically within < 1°C of the air temperature throughout the campaign, and hereafter, we refer to both temperatures as the same. The meteorological parameters measured in the SMEAR II Siikaneva 1 station for PAR, relative humidity, air temperature, the estimated leaf temperature, and the concentrations of the four measured terpene classes (isoprene, MTs, SQTs, DTs) are given in Fig. S6.

Isoprene showed the highest ambient mixing ratio among all the terpenes. We observed a relatively high ambient isoprene mixing ratio with a mean of 0.62 ± 0.8 ppbv during the campaign. A previous study at the same site during the same season reported a mean of 0.23 ± 0.009 ppbv (Haapanala et al., 2006). We observed a clear diurnal pattern in the ambient isoprene mixing ratio, reaching as high as 4.6 ppbv at 11:30 (UTC +2) on 22 June 2021. We observed a substantial accumulation of MTs, SQTs, and DTs during the night at Siikaneva, with MT concentrations reaching above 12 ppbv in the early morning on 11 June 2021. The highest SQT and DT mixing ratios were observed at 22:30 on 25 June 2021, at 1.22 ppbv and 23 pptv (parts per trillion by volume). We assume that these nighttime peak mixing ratios were due to the formation of a very shallow boundary layer in atmospherically stable conditions over the wetland at nighttime. A recent study also observed this phenomenon at the same site, and they have connected it to stronger NPF in the wetlands than in the surrounding boreal forests (Junninen et al., 2022). MT, SQT, and DT mixing ratios generally dropped during the daytime, presumably due to an increasing boundary layer height, ensuing dilution, and higher oxidant levels. The campaign-average mixing ratios were 1.1 ppbv for MT, 100 pptv for SQT, and 3 pptv for DTs. DTs have very low volatility and were thought not to be emitted by terrestrial vegetation for a long time. Li et al. (2020) report average mixing ratios of around 2 pptv of DTs in the French Landes forest during summertime. Fischer et al. (2021) measured DT mixing ratios of 0.1 pptv above a boreal forest.

### 3.2 Fluxes of terpenes from Siikaneva

As part of the EC flux analysis with the innFLUX code (Striednig et al. (2020), the data quality was investigated and grouped into nine quality classes (see SI section S1.3). For the following flux analysis, we have only used data that fulfills the criteria for quality classes 1-3 (Table S3) and with a minimum friction velocity of 0.1 m/s. Most of these data points were measured during the daytime. About 70 % of the available daytime data passed the filtering criteria. We had 496, 425, 483, and 449 data points of 30-minute flux time intervals for isoprene, MT, SQT, and DTs that passed the quality control.

We observed vertical fluxes for more than 250 out of 1072 peaks in our mass spectrometric data set. Besides terpenes, we also detected oxygenated VOCs, including smaller ones, such as methyl ethyl ketone ($C_4H_8O$) and methyl vinyl ketone ($C_4H_6O$), and compositions consistent with first oxidation products of sesquiterpenes (e.g., $C_{15}H_{24}O_3$, $C_{15}H_{24}O_4$). Earlier BVOC emission studies at Siikaneva were carried out by gas chromatography using chamber and relaxed eddy accumulation (REA) techniques. They reported 9 and 29 compounds each (Haapanala et al., 2006; Hellén et al., 2006), mostly $C_2$- to $C_{10}$-hydrocarbons. Many of these smaller organics have lower proton affinities than water and could not be detected using Vocus-PTR. Other studies using PTR-based measurement techniques at other locations have similarly detected vertical fluxes for 200 or more compounds (Fischer et al., 2021; Millet et al., 2018; Park et al., 2013). In the following, we focus our analysis on the emissions of the terpene compound groups.



Several studies have shown that isoprene is the main BVOC emitted by boreal wetlands (Klinger et al., 1994; Janson and De Serves, 1998). In this study, the highest emission fluxes were observed for isoprene: more than a factor of 6 times
higher than for MTs (Fig. 2b), on average. Overall, we observed a similar emission pattern for all terpenes. We also report the first-ever emission of DTs from wetlands (Fig. 2d). We observed relatively low emissions for all the terpenes from the start of the campaign till the end of May. During that period, the average temperature was below 10 °C, and the LAI (~0.2) was very low. We observed downward flux for MTs during nighttime from 1 to 12 June. However, most nighttime data did not pass the EC quality controls and could thus not be evaluated. DT fluxes were erratic at the beginning of the
campaign until the end of May since the measurements were close to LoD.

Overall, the terpene emissions increased substantially over the course of the campaign due to the increase in temperature and the progression of the growing season of the vegetation in the fen. LAI varied from 0.2 to 0.54, increasing continuously throughout the campaign from the beginning to the peak of the growing season (Fig. 2). The maximum flux for isoprene of 19.5 nmol m$^{-2}$ s$^{-1}$ was observed at noon (12:00-12:30) on 22 June 2021, and the maximum flux for MT of
2.2 nmol m$^{-2}$ s$^{-1}$ in the afternoon (14:30-15:00) of the same day, which was the day with the highest temperature during the campaign (31.7 °C at 15:00). During these maximum flux time intervals, the temperature was 30°C and 31.5°C, and PAR was 1533 and 1409 μmol m$^{-2}$ s$^{-1}$, respectively. The maximum emission for SQTs of 1.25 nmol m$^{-2}$ s$^{-1}$ was observed already on 21$^{st}$ June (10:00-10:30) at 27.9°C and 1470 μmol m$^{-2}$ s$^{-1}$ of PAR, and the maximum emission for DTs of 0.018 nmol m$^{-2}$ s$^{-1}$ was observed on 22$^{nd}$ June (10:30-11:00) at 29.5°C and 1435 μmol m$^{-2}$ s$^{-1}$ of PAR. Figure S7 shows the
diurnal patterns of the terpene fluxes with the campaign-wide median values and the interquartile ranges (IQR). The average fluxes observed during the day (PAR>50 μmol m$^{-2}$ s$^{-1}$) were 1.96 nmol m$^{-2}$ s$^{-1}$ for isoprene, 0.33 nmol m$^{-2}$ s$^{-1}$ for MTs, 0.2 nmol m$^{-2}$ s$^{-1}$ for SQTs, and 0.003 nmol m$^{-2}$ s$^{-1}$ for DTs. The maximum nighttime deposition flux observed for MTs was -0.11 nmol m$^{-2}$ s$^{-1}$. All other terpenes had one order of magnitude lower deposition fluxes.

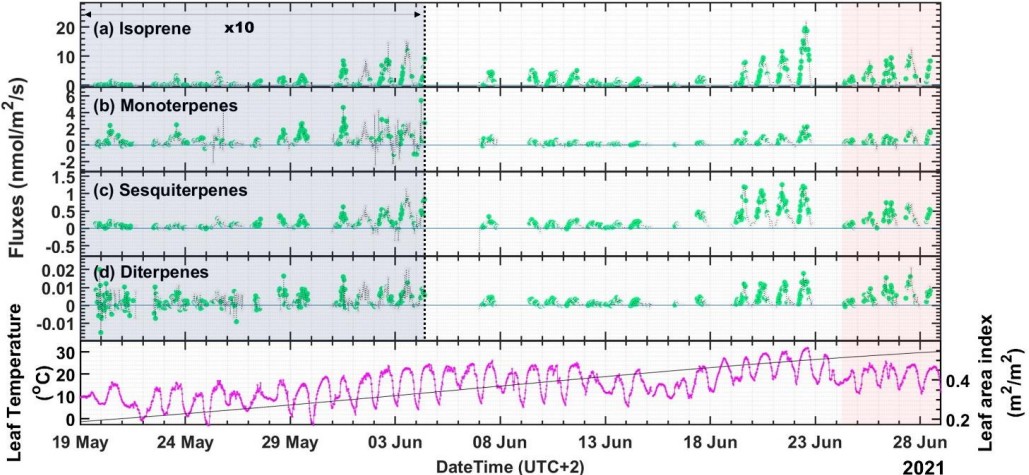

**Figure 2**. Fluxes of terpenes (a-d) and leaf temperature with leaf area index (LAI) (bottom panel) measured throughout the campaign. Note that the panels (a)-(d) are split into two, and the flux measurements up to 4 June are multiplied by 10 to increase the visibility of the lower values earlier in the campaign. Fluxes colored in green (class 1-3) are good flux periods and are used for fundamental research. The gray data points can only be used for orientation of flux or other





general purposes. The red shaded area is the period after the maximum temperature day, 22nd June 2021, when the highest
air temperature above 31 °C was recorded and is referred to as the "high-temperature stress period" in the manuscript.

Previous studies at the same site have measured isoprene along with other small hydrocarbons, namely C2-C5 alkanes
and halocarbons (Haapanala et al., 2006; Hellén et al., 2006). The previous REA study at the site reported a maximum of
3 nmol m$^{-2}$ s$^{-1}$ for isoprene on 4th August 2004 at 25 °C and 1500 µmol m$^{-2}$ s$^{-1}$ PAR (Fig 3 in (Haapanala et al., 2006)).
The other previous study used enclosure measurements and reported a much lower isoprene emission flux of 0.4 nmol m$^{-2}$ s$^{-1}$, on 18 August 2005 at 25 °C temperature and 1135 µmol m$^{-2}$ s$^{-1}$ PAR. That enclosure study suggested that *Sphagnum*
moss could be a source of the isoprene emitted by the fen (Hellén et al., 2006). Both studies have not reported emission
fluxes of any terpenes other than isoprene from this site.

In our study, SQTs emitted from the boreal fen had almost of the same magnitude as the emitted MTs. In contrast, a recent
study in a subarctic fen in northern Finland (Hellén et al., 2020) reported that SQT emissions were significantly higher
than MT emissions. Their highest SQT emission was 0.012 nmol m$^{-2}$ s$^{-1}$, and MTs had a flux of 0.004 nmol m$^{-2}$ s$^{-1}$ at 20
°C temperature and 560 µmol m$^{-2}$ s$^{-1}$ PAR, which is much lower than our observations.

Only a few ecosystem-scale EC/disjunct EC studies have been conducted in similar high latitude environments, two in a
Swedish subarctic fen (Seco et al., 2020; Holst et al., 2010) and one in an Alaskan tundra ecosystem (Potosnak et al.,
2013). A recent study by Seco et al. (2020) at a subarctic mire in Abisko, Sweden (68°20' N, 19°03' E), reports maximum
isoprene emission of 11.3 nmol m$^{-2}$ s$^{-1}$ at 21.2°C air temperature and 1339 µmol m$^{-2}$ s$^{-1}$ of PAR on 16 July 2018 which is
about the same magnitude we observed in our study. They also report a maximum MT emission flux of 0.2 nmol m$^{-2}$ s$^{-1}$
at 19.6°C air temperature and 1145 µmol m$^{-2}$ s$^{-1}$ of PAR which is more than an order of magnitude less than the highest
monoterpene emission observed in our study. During their study, they had leaf temperature measurements and had much
higher leaf temperatures than air temperatures. The setup we used for our canopy environment model was able to predict
their leaf temperature from their air temperatures. We believe the higher leaf temperatures for their campaign could be
due to high PAR, which heats the leaf surface, but the air temperature remains lower due to large-scale meteorological
reasons. Seco et al. (2020) could not detect any ecosystem-scale SQT or DT emissions due to the limitations of their
measurement setup. Another disjunct EC study (Holst et al., 2010) at the same Abisko mire site as Seco et al. (2020)
reports a maximum isoprene emission flux of 6.4 nmol m$^{-2}$ s$^{-1}$ at 22°C air temperature and 1200 µmol m$^{-2}$ s$^{-1}$ of PAR on
3 August 2006. The Alaskan study was conducted in a moist tundra ecosystem (68°38' N, 149°38' W) (Potosnak et al.,
2013) and reports a maximum hourly isoprene emission flux of 5.5 nmol m$^{-2}$ s$^{-1}$, observed at an air temperature of 22 °C
and a PAR level over 1500 µmol m$^{-2}$ s$^{-1}$. These isoprene emissions are 3-4 times lower than the maximum isoprene
emission flux observed in our study but have at least 10 °C lower temperatures than observed in our campaign. We
observed isoprene emissions of 2-10 nmol m$^{-2}$ s$^{-1}$ when temperatures and PAR were comparable. Moreover, even though
these studies were conducted in similar high latitude environments with low species diversity, they have different plant
species mixes, and we might expect different emissions. The Swedish mire has vegetation similar to our site in its wet
areas, whereas the vegetation in the drier area is different. In the Alaskan study, the vegetation is dominated by different
species such as *Salix* spp. This also points to the need for more than one flux study in each ecosystem.
Several chamber studies report isoprene fluxes from boreal fens, starting with Janson and De Serves (1998) in central
Sweden (Stormossen, 60°10'N, 17°05'E). Using a static chamber technique, they report isoprene fluxes up to 9.3 nmol
m$^{-2}$ s$^{-1}$ during August. A study by (Bäckstrand et al., 2008) reported an average flux of non-speciated total non-methane





volatile organic compound (NMVOC) flux of 140 mg C m$^{-2}$ d$^{-1}$, which would convert to 27 nmol m$^{-2}$ s$^{-1}$ assuming all the NMVOC was isoprene. This is about the same magnitude as our observed maximum fluxes.

No other study has ever reported ecosystem-scale DT emissions, except for Fischer et al. (2021). They used a similar setup as ours in Hyytiälä, above a boreal forest canopy close to our site. They found an average emission of 0.15 pmol m$^{-2}$ s$^{-1}$, which is more than an order of magnitude lower than the DT emissions we measured here. Measuring accurate emissions of the larger terpenes, such as SQT and DTs, is very important since very little of their emission potential is known. Therefore, their impact on CCN growth is uncertain. A recent study has found boreal wetlands to initiate stronger

NPF than boreal forests (Junninen et al., 2022). We speculate that the unexpectedly high emissions of SQT and DTs could contribute to the strong NPF observed there.

### 3.3 Emission modeling

#### 3.3.1 Emission factors of terpenes

We used Eq 2-10 and calculated $C_L$ and $C_T$ activity factors for G93 and G2012 emission activity algorithms using the leaf

temperature and PAR for each half-hour flux data point. For the G93, $C_L$ and $C_T$ activity factors are based on the instantaneous response of emissions to leaf temperature and PAR. For the G2012 algorithm, $C_L$ and $C_T$ activity factors are based on instantaneous, the past one-day and 10-day average leaf temperatures and PAR. Using Eq 1, we can derive the emission factor at standard conditions (EF$_i$ at T=30 °C, PAR=1000 μmol m$^{-2}$ s$^{-1}$). Figure S8 shows the quality-filtered, half-hourly isoprene fluxes plotted against the combined light ($C_L$) and temperature ($C_T$) activity factors ($C_L$ x $C_T$), panels

a-b, and $C_L$ x $C_T$ x LAI (panels c-d), respectively, as calculated in G93 (left panels) and G2012 (right panels). As per Eq. 1, the slope of the line fitted in panels c-d gives an estimate of EF for isoprene (EF$_{iso}$). The slopes in panels a-b correspond to EF$_{iso}$, where the influence of LAI is not considered (i.e., LAI=1).

If we neglect LAI (i.e., LAI=1) in Eq 1, we obtained EF$_{iso}$ of 11.1±0.3 nmol m$^{-2}$ s$^{-1}$ (R$^2$=0.77) using the G93 algorithm and EF$_{iso}$ of 5.3±0.1 nmol m$^{-2}$ s$^{-1}$ (R$^2$=0.8) using the G2012 algorithm. The maximum LAI of the fen is around 0.5, and

all previous fen BVOC studies neglect LAI and assume it to be 1 when calculating EF. Earlier studies from the same site have reported EF$_{iso}$ using the G93 algorithm of 0.91 and 2.77 nmol m$^{-2}$ s$^{-1}$ (Haapanala et al., 2006; Hellén et al., 2006), about 5-10 times lower than our estimate. A recent study by Seco et al. (2020) and another study by Holst et al. (2010) at Abisko mire reported EF$_{iso}$ using the G93 algorithm of 5.8 nmol m$^{-2}$ s$^{-1}$ and 5.32 nmol m$^{-2}$ s$^{-1}$, respectively, which is two times lower than our estimate. If we include LAI in Eq 1., we obtain EF$_{iso}$ of 22.7±0.5 nmol m$^{-2}$ s$^{-1}$ (R$^2$=0.82) for the G93

algorithm and EF$_{iso}$ of 10.9±0.2 nmol m$^{-2}$ s$^{-1}$ (R$^2$=0.84) using the G2012 algorithm.

We also split our dataset into two subsets to fit EF for each of them separately instead of a single EF: one for $C_L$ x $C_T$ < 0.5 (EFlow$_{iso}$) and one for $C_L$ x $C_T$ > 0.5 (EFhigh$_{iso}$). We observe that EFlow$_{iso}$ << EFhigh$_{iso}$, with EFlow$_{iso}$ values more compatible with the results from the previous studies. An interpretation of this finding is that the high light and temperature conditions leading to $C_L$ x $C_T$ > 0.5, which are higher than typical values for boreal wetlands, would increase

isoprene emissions more than G93 and G2012 predict.

**Table 1.** Emission factor (EF) at standardized conditions for terpenes using G93 algorithm (LAI =1), G2012 algorithm (actual LAI), using Eq. 12 with the temperature coefficient β, MEGANv2.1 (Arctic C3 grass) and in literature.

| Terpene | EF (G93, LAI=1) (nmol m⁻² s⁻¹) | EF (G2012, actual LAI) (nmol m⁻² s⁻¹) | Eq. 12 EF (nmol m⁻² s⁻¹) | β | MEGANv2.1 (nmol m⁻² s⁻¹) Arctic C3 grass | Literature (nmol m⁻² s⁻¹) |
|---|---|---|---|---|---|---|
| **Isoprene** | 11.1 | 10.9 | 12 | 0.23 | 6.3 | 0.91-5.8 |
| **MT** | 1.2 | 2.4 | 1.2 | 0.18 | 0.025 | - |
| **SQT** | - | 1.3 | 0.9 | 0.17 | 0.008 | - |
| **DT** | - | - | 0.01 | 0.18 | - | - |


As described in the methods section, the G93 algorithm only uses temperature to predict monoterpene emissions and has no emission algorithm for sesquiterpenes. The G2012 algorithm includes temperature and light-dependent emissions for MT and SQT (section 2.5). As for isoprene (Fig. S8 right), we used G2012 to estimate the emission factors of MTs and SQTs in standard conditions ($EF_{MT}$ and $EF_{SQT}$, T=30 °C, PAR=1000 μmol m⁻² s⁻¹). Figure S9 shows the quality-filtered

half-hourly MT (left panels) and SQT (right panels) fluxes plotted against the combined $C_L$ x $C_T$ factors (panels a-b) and $C_L$ x $C_T$ x LAI (panels c-d), respectively. We obtained an $EF_{MT}$ of 0.83 ±0.03 nmol m⁻² s⁻¹ ($R^2$=0.62) and an $EF_{SQT}$ of 0.63±0.02 nmol m⁻² s⁻¹ ($R^2$=0.72). When including the influence of LAI, we obtained an $EF_{MT}$ of 2.4±0.08 nmol m⁻² s⁻¹ ($R^2$=0.66) and an $EF_{SQT}$ of 1.3±0.03 nmol m⁻² s⁻¹ ($R^2$=0.79). Table 1 shows the EF for isoprene, MT, and SQT using the G93 algorithm excluding LAI and the G2012 algorithm including LAI.

The red color-coded data points in Figs. S8, S9, 3, and 4 were measured after 22 June when the highest air temperatures above 31 °C had been recorded, corresponding to the red shade in Fig. 2. We can clearly see that the red points are underpredicted by the G93 algorithm based on instantaneous temperature response (Fig. S8(a) and (c)). This shows that the abnormally high temperature experienced by the vegetation on 22 June can induce elevated emissions afterward. Hence, we refer to this period as the "high-temperature stress period." A similar dependence of emissions on the past

temperature in a fen was also observed by Ekberg et al. (2009). The G2012 algorithm predicts isoprene emissions better than the G93 algorithm. However, $C_T$ values for the high-temperature stress were still underpredicted by G2012. We, therefore, look more deeply into the temperature dependence of the emissions and compare it to standard BVOC models.

### 3.3.2 Temperature dependence of terpene emissions

For the following analysis, we chose the data points saturated by light (PAR > 1000 μmol m⁻² s⁻¹) to ensure that the

variability in emissions was only due to temperature fluctuations. Figure 3 shows the factor by which terpene emissions increase for every 10°C rise in temperature ($Q_{10}$ values) calculated by fitting the terpene flux data with the log-linear equation (Seco et al., 2020):

$$\log (E_i) = T \frac{\log Q_{10}}{10} + a_0 \quad (11)$$

For comparison, the dashed black lines in Fig. 3 show the temperature dependence of the temperature activity factor ($C_T$)

of the G93 model, scaled for the emissions at 30 °C. Since we observed leaf temperatures below 32 °C and did not observe a temperature optimum for the emissions of terpenes, we also fit the data with the light-independent temperature response used for MT of the G93 algorithm (i.e., modification of Eq 1):





$$E_i = EF_i \exp^{\beta (T-303)} \quad (12)$$

In Figure 3, the fitted temperature coefficient $\beta$ is shown, and the fitted curve is shown in magenta. We did not include

the red color-coded data for the fit (the flux measurements after the maximum temperature day, 22 June) since they did
not follow the G93 algorithm. The obtained EF and $\beta$ values are given in Table 1.

Our fitted $Q_{10}$ and $\beta$ values were much higher than those used for these parameters in the MEGAN G93 emission activity
algorithm. G93 uses a $Q_{10}$ of 3.3 for isoprene and 2.5 for MTs, and a $\beta$ of 0.13 for isoprene and 0.1 for most MTs. The
typical values of $Q_{10}$ used in most emission models vary between 3 and 6 for isoprene (Peñuelas and Staudt, 2010). Here,

we obtained $Q_{10}$ values of 16.9 for isoprene ($R^2$=0.9), 4.9 for MT ($R^2$=0.7), 12.5 for SQT ($R^2$=0.8), and 7.85 for DTs
($R^2$=0.7). These $Q_{10}$ values are up to a factor of 5 higher than those currently used in the G93 algorithm. A recent study
by (Seco et al., 2020) also found significantly higher $Q_{10}$ values for isoprene. They derived $Q_{10} = 131$ using air temperature
and $Q_{10} = 14.5$ using leaf temperature. Note that the leaf temperature was generally much higher than the air temperature
in that study, whereas we had minimal differences here. Our $Q_{10}$ for isoprene is similar to that calculated by Seco et al.

(2020) using leaf temperature. Studies that artificially warmed the vegetation in tundra ecosystems in Northern Sweden
and Greenland have shown $Q_{10}$ values of 10 and 22 (Tang et al., 2016; Kramshøj et al., 2016), which is higher than the
G93 algorithm and closer to our result. Our high $Q_{10}$ values for all the terpenes show that boreal wetland vegetation has
high sensitivity towards increasing temperature. Overall, rising temperatures will therefore lead to even higher emissions
than predicted by commonly used BVOC emission models (Guenther et al., 2012).

A closer look at the isoprene behavior shown in Fig. 2 and 3 suggests that there may be a threshold after which higher
emissions "turn on." The threshold could be a specific temperature but more likely a period of elevated temperature days.
The red points in Fig. 3, which we think are after hitting the threshold, have a lower decrease in isoprene emissions when
the temperature drops from 30 °C to 22 °C and broadly follow the temperature dependence of the G93 model. Our
observations fit with the observations reported by Monson et al. (1994) for a different vegetation type. They observed

that leaves that emerged during the cool spring induced higher isoprene emissions when exposed to higher temperatures.
This shows the inadequacy of using one single EF throughout the growing season. To get an accurate estimate of
emissions from this ecosystem, the emissions during the start of the growing season should be predicted with one EF with
strong temperature dependence. Once the threshold is reached, a higher EF along with the G93 temperature dependence
could give an accurate estimate.




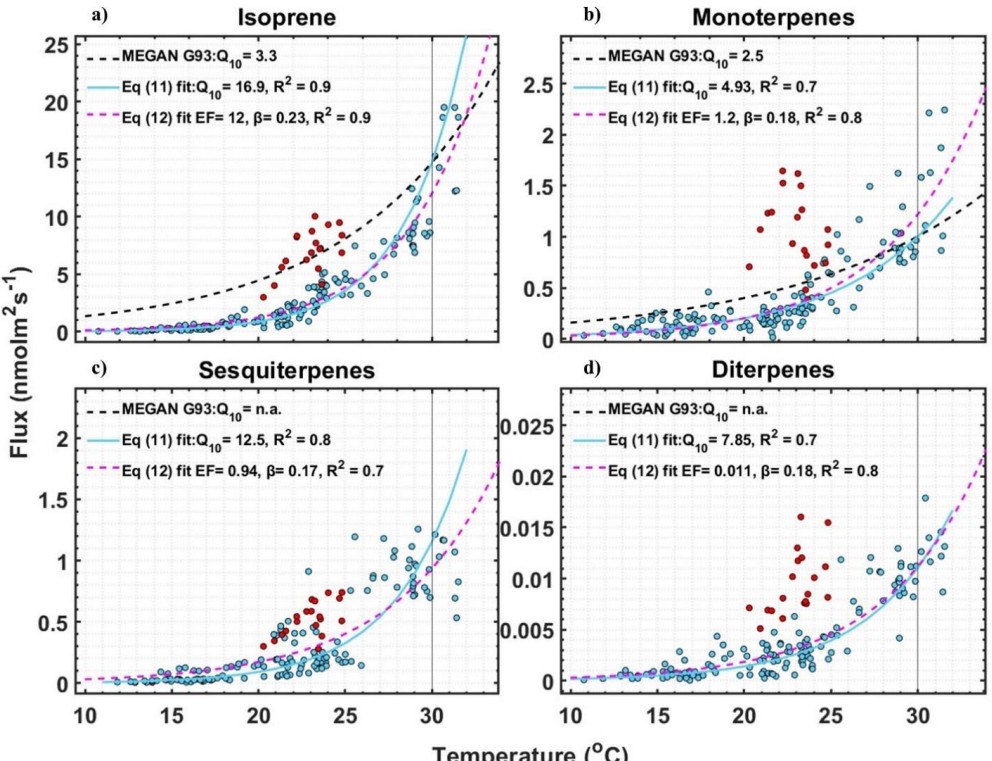

**Figure 3**. Measured fluxes of (a) isoprene, (b) monoterpene, (c) sesquiterpene, and (d) diterpenes fluxes vs. temperature of data points that passed EC quality criteria and that have PAR > 1000 µmol m$^{-2}$ s$^{-1}$. $Q_{10}$ values were calculated by fitting the data with Eq 11. EF and β are calculated by fitting the data with Eq 12. The dashed black line shows the relationship with leaf temperature of the temperature activity factor ($C_T$) of the G93 model (Guenther et al., 1993), scaled to match emissions at 30°C. The red data points are the flux measurements after the maximum temperature day, 22nd June 2021, when the air temperature crossed 31°C and are not included in the fits.

### 3.3.3 Comparison to MEGANv2.1

It is clear from Table 2 and Figure 3 that terpenes have higher emissions than in previous studies, and they do not follow the standard temperature dependence as used in the G93 algorithm. Hence, we compare our emission measurements with model predictions of the MEGANv2.1 emission model for a single measurement site using its EF for the Arctic C3 grass plant functional type (Guenther et al., 2012) given in Table 1. The EF (in µg m$^{-2}$ hr$^{-1}$) is 1600 for isoprene, 12.5 for MT, and 6 for SQT. Figure 4a-c shows the measured vs. modeled emission for isoprene, MTs, and SQTs. The fitted slopes give the factors by which the emissions are underestimated. The model underestimates emissions by over 9 times for isoprene, 340 times for MTs, and 800 times for SQTs. If we use these factors to correct the EF for Arctic C3 grass, the corrected EF$_{iso}$ would be as high as broadleaf trees and shrub functional types (50 nmol m$^{-2}$ s$^{-1}$), and the corrected EF$_{MT}$ and EF$_{SQT}$ would be higher than needle leaf and broadleaf trees functional types (2-3 nmol m$^{-2}$ s$^{-1}$) (Guenther et al., 2012). The corrected EFs are of the same magnitude as the EF$_{iso}$, EF$_{MT,}$ and EF$_{SQT}$ obtained in our study using Eq. 2-5 (Table 1).



We also compared the temperature response ($C_T$) of the measured emissions for isoprene, MTs, and SQTs with the $C_T$ of the MEGANv2.1 emission model. We derived the measured $C_T$ by dividing the measured emissions when PAR was saturated (>1000 μmol m$^{-2}$s$^{-1}$) by the EF obtained using Eq. 12. The modeled $C_T$ for the same data points was calculated using Eq 5. We then calculate the residual $C_T$ (measured $C_T$ – MEGANv2.1 $C_T$), and Fig. 4d-f shows this residual $C_T$ for isoprene, MT, and SQT vs. leaf temperature. From Fig. 4d-f, we see that the temperature response in the MEGAN model

may sufficiently represent the temperature dependence of our observations up to around 20 °C but falls short at higher temperatures for all three terpenes represented in the model. The residuals are highest for MT due to the higher MT emissions during the "high-temperature stress" period. This direct comparison with single point MEGANv2.1 shows that there is drastic underestimation in the EF and also temperature dependence of the model for this vegetation type. These underestimations of emissions by the model could significantly impact modeling the atmospheric chemistry in boreal

environments.

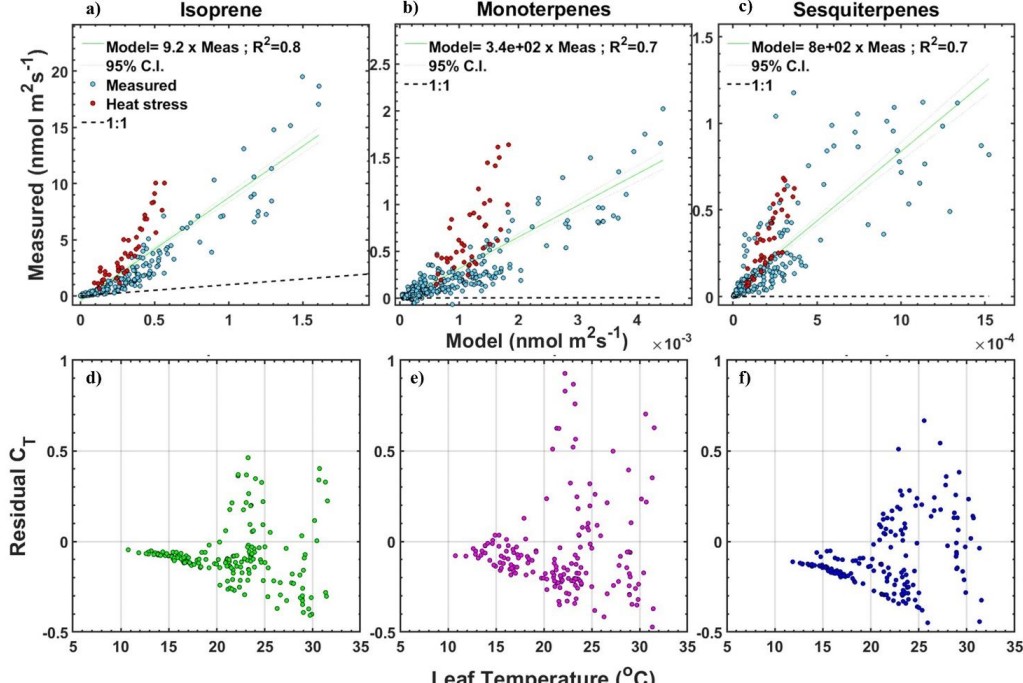

**Figure 4**. Measured vs. MEGANv2.1 emission for (a) isoprene, (b) MTs, and (c) SQTs of data points that passed EC quality criteria. The red color-coded data points are the flux measurements after 22nd June 2021 and are included in the fits. Residual $C_T$ (measured $C_T$ – MEGANv2.1 $C_T$) of (d) isoprene, (e) MTs, and (f) SQTs vs. leaf temperature of data

points that passed EC quality criteria and that have PAR > 1000 μmol m$^{-2}$s$^{-1}$ for the whole campaign.

**4 Summary and conclusions**

In this study, we present ecosystem-scale eddy covariance (EC) fluxes of the four classes of terpenes, namely isoprene, monoterpenes (MTs), sesquiterpenes (SQTs), and diterpenes (DTs), which globally make up over 70% of all BVOCs emitted from terrestrial vegetation. The measurements were conducted at the Siikaneva boreal fen from the start to the


peak of the growing season of 2021 (mid-May/late spring to the end of June/mid-summer). These are the first EC fluxes
of terpenes to be reported based on measurements using the novel state-of-the-art Vocus-PTR mass spectrometry
technique and the first-ever fluxes for DTs from a wetland. Using this instrument's improved sensitivity, compared to
classic PTR mass spectrometers, and a new inlet to reduce vapor-surface interactions during sampling, we detected fluxes
for about 250 compounds from C1 to C20 hydrocarbons and oxygenated organics.

As found by previous studies, isoprene was the dominant terpene emitted by the wetland, followed by MTs, SQTs, and
DTs. We calculated the emission factor in standard conditions (T=30 °C, PAR=1000 μmol m$^{-2}$ s$^{-1}$, and LAI =1) of the fen
using the MEGAN model with the G93 emission activity algorithm (Guenther et al., 1993) for isoprene (EF$_{iso}$) and MTs
(EF$_{MT}$) and the simplified MEGANv2.1-G2012 emission activity algorithm (Guenther et al., 2012) for isoprene, MTs,
and SQTs (EF$_{SQT}$). We also used the light-independent temperature response for MT of the G93 algorithm, which is an
exponential function to calculate the EF of all terpenes for light saturated flux data points. We estimate an EF$_{iso}$ of 11 ±
0.3 nmol m$^{-2}$ s$^{-1}$, which is at least two times higher than previous studies (Haapanala et al., 2006; Seco et al., 2020) and as
high as the emission factors typical for broadleaf trees in the lower latitudes. EF$_{MT}$ and EF$_{SQT}$ were 2.4± 0.1 and 1.3± 0.03
nmol m$^{-2}$ s$^{-1}$, higher than needle leaf and broadleaf tree functional types (Guenther et al., 2012). We observed an EF for
DTs of 0.011 nmol m$^{-2}$ s$^{-1}$, which is the first estimate for this ecosystem.  During the campaign, the temperature peaked
above 31°C, which is abnormally high for boreal environments, and we observed that the terpene emissions remained
elevated even though the temperatures declined. We denoted this period as the "high-temperature stress period," and the
maximum flux for all the terpenes coincided with this period. The emissions peaked at 19.5 nmol m$^{-2}$ s$^{-1}$ for isoprene, 2.2
nmol m$^{-2}$ s$^{-1}$ for MT, 1.25 nmol m$^{-2}$ s$^{-1}$ for SQT, and 0.018 nmol m$^{-2}$ s$^{-1}$ for DTs. The maximum isoprene emission flux is
significantly higher (by a factor of 2-15) than in all previous studies conducted in similar ecosystems (Haapanala et al.,
2006; Seco et al., 2020). However, the maximum temperatures in these previous studies were also much lower (by 7-
10°C).

We also note that the treatment of LAI is an important aspect when predicting terpene emissions. The satellite-based LAI
values for June (Copernicus 300m LAI) for Siikaneva were around 1.7 (https://land.copernicus.eu/global/products/lai),
whereas the site-specific LAI based on in-situ measurements was only 0.37 on average and is more representative and
accurate. Therefore, using the EF derived in this study together with satellite-based LAI might overestimate the emissions
by 3-5 times, for example, while running MEGAN in a regional/global model. This indicates a need for a different
approach for "standardizing" landscape average fluxes when there is no tree canopy. Getting the LAI right (for instance,
whether 0.5 or 5) is essential, and the approach used for forest canopies may not be appropriate for such environments.

Even though the terpene emissions showed an exponential temperature dependence, the G93 algorithm did not adequately
represent the high-temperature stress period since this parametrization is based only on instantaneous light and
temperature. On the other hand, the G2012 algorithm also considers the history of light and temperature and their
instantaneous counterparts and estimates the emissions better. However, the terpene emissions during the "high-
temperature stress period" are still underestimated also by the G2012 algorithm. To further understand the sensitivity of
the terpene emissions to temperature, we calculated the Q$_{10}$ values, the factor by which terpene emissions increase for
every 10 °C rise in temperature. We obtained Q$_{10}$ values up to 5 times higher than those widely used in BVOC emission
models but similar to recent studies in the arctic and subarctic that measured Q$_{10}$ values for isoprene (Seco et al., 2020;
Holst et al., 2010; Tang et al., 2016).

We compared our measured emission of terpenes with the MEGANv2.1 emission model, specifically using EF for the
"C3 Arctic grass" plant functional type. We found the emissions were underestimated by over 9 times for isoprene, over





300 times for MTs, and 800 times for SQTs. We also found that MEGANv2.1 agreed with the measured temperature response only up to 20 °C and failed to reproduce the measured temperature dependence for higher temperatures. These findings of both high EF and high sensitivity to increasing temperatures imply that these high latitude ecosystems have the potential to be a significant source of terpenes, especially isoprene, into the atmosphere, in particular in a warming climate.

Future studies may gain valuable insights by using our observations of terpene emissions from wetland ecosystems. Our results illustrate that applying the default EF (C3 Arctic grass) and temperature parameterizations in MEGANv2.1 can lead to substantial inaccuracies in modeling terpene emissions. The parametrizations were developed mainly based on temperate and tropical ecosystems. We hypothesize that the much stronger temperature response seen in our measured terpene fluxes from the boreal fen could be due to the high latitude vegetation being acclimatized to colder environments

and particularly susceptible to heat stress.

In light of our results, we find that there may be too few studies of BVOC (terpene) emissions from these high latitude ecosystems. In particular, longer-term measurements of BVOCs are lacking for all ecosystems. Our study demonstrates how current, highly sensitive mass spectrometers, like Vocus-PTR, can be used in conjunction with EC calculations to measure ecosystem-scale VOC fluxes accurately, even for larger terpenes, and without disturbing or stressing the

environment. Such measurements appear particularly important with our finding that terpene emissions from boreal wetlands are very sensitive to temperature change. Conceivably, anthropogenic global warming can induce much higher BVOC emissions in the future. More studies will be necessary to understand the steep temperature dependence of terpene emissions in wetland ecosystems, including potential differences between specific wetland types and the underlying physiological mechanisms. Also, the data presented here contains only the period until the peak of the growing season.

Longer-term studies covering multiple seasons, better yet years, may be warranted. It should also be kept in mind that longer-term ecosystem responses to climatic changes will affect BVOC emissions.

**5 Code and Data availability**

The terpene flux data and other parameters used in this article are available for download at 10.5281/zenodo.7002511 (Vettikkat et al., 2022). Analysis code is available from the corresponding authors upon request.

**6 Author contributions**

S.Scho., L.V., and P.M. conceived and designed the study. L.V. carried out this work as part of his Ph.D. thesis under the supervision of S.Scho. L.V. performed Vocus-PTR measurements, collected GC adsorbent tubes, carried out all analyses, interpreted the data, and wrote the manuscript. A.B., and P.R., helped in troubleshooting the Vocus-PTR. P.R. and S. Scha. helped with calibration analysis. R.S. and A.B.G. helped in interpreting the results and comparing the measured

data with MEGAN. H.Y. analyzed the GC adsorbent tubes. S.Scho. revised the paper and supervised all experimental and analysis aspects. E.M. and E.S.T. provided the LAI and information about vegetation at Siikaneva. All authors contributed to the final draft.

**7 Competing interests**

The authors have no competing interests to declare.



**8 Acknowledgments**

We thank Heidi Hellén, Pontus Roldin, Robin Wollesen de Jonge, Lukas Fischer, Michael Boy, and Petri Clusius for valuable discussions. We highly appreciate the help and support provided by the SMEAR II station team, especially Matti Salminen and Lauri Ahonen, in setting up and maintaining our measurements in Siikaneva. Lejish Vettikkat acknowledges the UEF EPHB doctoral degree programme for funding. This work was financially supported by the Academy of Finland Flagship programme (grant no. 337550), Academy of Finland project no. 310682. Simon Schallhart acknowledges Academy of Finland project no. 323255. Roger Seco acknowledges a Ramón y Cajal grant (RYC2020-029216-I) funded by MCIN/AEI/ 10.13039/501100011033 and by "ESF Investing in your future". IDAEA-CSIC is a Severo Ochoa Centre of Research Excellence (MCIN/AEI, Project CEX2018-000794-S).

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
