# Peer review of "High emission rates and strong temperature response make boreal wetlands a large source of isoprene and terpenes"

_Atmospheric Chemistry and Physics, 2022_

## Author Comment (AC1)

Please find the revisions/replies (in blue) to the specific points (in black) raised by the reviewers and the changes made in the manuscript (in green) for easy perusal.

**Reply to comments by RC1 (received 1 November 2022)**

**General comments:** The manuscript by Vettikkat et al. presents an impressive EC BVOC dataset focusing on terpenes in a Boreal wetland. It is well written and publication is warranted after addressing some minor comments below.

Reply: We thank the reviewer for their encouraging and valuable remarks that the presented EC dataset is impressive and for deeming that the publication of the manuscript is warranted after addressing the minor comments raised in this review.

 **Specific comments:**

1. The abstract is too long. A large part of the abstract reads more like an Introduction. The abstract could be shortened to highlight the main findings more concisely.

Reply: We agree with the reviewer that the abstract is lengthy and needed to be more concise. We have edited the Abstract as follows:

Wetlands cover only 3 % of the global land surface area but boreal wetlands are experiencing unprecedented warming of four times the global average. These wetlands emit isoprene and terpenes (including monoterpenes (MT), sesquiterpenes (SQT), and diterpenes (DT)), which are climate-relevant highly reactive biogenic volatile organic compounds (BVOCs) with an exponential dependence on temperature. In this study, we present ecosystem-scale eddy covariance (EC) fluxes of isoprene, MT, SQT, and DT (hereafter referred to together as terpenes) from Siikaneva, a boreal fen in southern Finland, from the start to the peak of the growing season of 2021 (19 May 2021 to 28 June 2021). These are the first EC fluxes reported using the novel state-of-the-art Vocus- proton transfer reaction mass spectrometer (Vocus-PTR) and the first-ever fluxes reported for DTs from a wetland. Isoprene was the dominant compound emitted by the wetland, followed by MTs, SQTs, and DTs, and they all exhibited a strong exponential temperature dependence. The $Q_{10}$ values, the factor by which emissions increase for every 10 °C rise in temperature, were up to 5 times higher than those used in most BVOC models. During the campaign, the air temperature peaked above 31 °C on 21-22 June 2021, which is abnormally high for boreal environments, and the maximum flux for all terpenes coincided with this period. We observed that emissions were elevated after this abnormally "high-temperature stress period," indicating that past temperatures alter emissions significantly. The standardized emission factor (EF) of the fen for isoprene ($EF_{iso}$) was $11.1 \pm 0.3$ nmol m$^{-2}$ s$^{-1}$, which is at least two times higher than in previous studies and as high as the emission factors typical for broadleaf and other forests in the lower latitudes. We observed $EF_{MT}$ of $2.4 \pm 0.1$ nmol m$^{-2}$ s$^{-1}$, $EF_{SQT}$ of $1.3 \pm 0.03$ nmol m$^{-2}$ s$^{-1}$, higher than typical for needle leaf and broadleaf tree functional types, and $EF_{DT}$ of $0.011 \pm 0.001$ nmol m$^{-2}$ s$^{-1}$. We also compared the landscape average emissions to the Model of Emissions of Gases and Aerosols from Nature (MEGAN) v2.1, and found that the emissions were underestimated by over 9 times for isoprene, over 300 times for MTs, and 800 times for SQTs. Our results show that due to very high EFs and high sensitivity to increasing temperatures, these high-latitude ecosystems can be a large source of terpenes to the atmosphere, and anthropogenic global warming could induce much higher BVOC emissions from wetlands in the future.

2. Line 108: What is a classical PTR-MS?

Reply: Here we wanted to make a distinction between the new generation Vocus-PTR and the previous generation of PTR-MS. The key differences being high sensitivity and lack of humidity dependence in the Vocus-PTR. We have changed the term 'classical' to 'previous-generation'.

3. Line 185 and Fig S5. VOCUS seems to underpredict isoprene or the GC seems to overpredict isoprene– Could this perhaps be related to a transmission issue/variation of VOCUS during the campaign for low

molecular weights – how was the transmission checked? For such a local isoprene source, one would expect that the flux stationarity criterium for isoprene (and others?) would be violated for a large fraction of the dataset (?). This does not seem to be the case. Alternatively the GC measurements could be overestimating isoprene concentrations. Were the GC cartridges cross calibrated on site with the isoprene standard?

Reply: We believe that the GC measurements are higher than the Vocus ones due to the difference in sampling location, i.e., the distance between the isoprene source on the ground and the sampling inlets. The Vocus samples air 2.4 m above the ground while the sample for GC analysis were collected at 2 m above the ground. The 10 Hz timeseries of Vocus indeed shows that the fluctuation in concentrations were drastic but did not violate the stationarity test. Figure R1 shows the isoprene timeseries with 10 Hz, 1 min, and 30 min time resolution. Pure isoprene standards were used to quantify the GC cartridges collected at the measurement site. Since the calibration of Vocus was performed routinely, calibration factors will take any changes including the transmission into account and hence no further transmission corrections were applied.

[Figure]

Figure R1. Isoprene time series with 10 Hz, 1 min, and 30 min time resolution in ppbv on 22.06.2021, the day with highest emission.

4. Figure 2: adding the x 10 only to the isoprene panel is somewhat confusing at first sight. Only after reading the figure caption it becomes clear that all compounds were multiplied by 10 for the first part of the campaign.

Reply: Thank you for pointing out that the meaning of "x10" was not clear at first glance. We have edited the Figure 2 to have the 'x10' at the top of the figure and use a darker color to make the area of the graph affected by the "x10" properly visible.

[Figure]

5. Figure 4 cc: The reported fluxes are extremely high. Higher than in most Boral forests, particularly also for DT. MT / SQT ratios have typically been reported as a 10 : 1 ratio in Boreal forests. Here, the emissions are comparable (e.g. 1:1) DT seems 50x higher than fluxes reported over a Boreal forest.

Reply: The fluxes observed are indeed high and the MT/SQT ratio is comparable. Recent study by Hellén et al. (2020) have shown that SQTs emissions from a fen can be even higher than MTs. The boreal wetland emissions are very different from boreal forests emissions and hence the comparison of the two different ecosystems would not be straightforward. Furthermore, this is the first study quantifying the emission of DTs from the wetland ecosystem as we do not have any other studies.

6. Line 535. The authors emphasize that such high fluxes could have significant ramification for studying NPF in the Boreal region and future studies should put more emphasis on wetlands. To that end, can the authors at least qualitatively estimate how much wetlands could contribute to reactive terpene emissions compared to Boreal forests (e.g. by scaling up based on land cover).

Reply: We thank the reviewer for this good suggestion. We add the following text to the manuscript Section 3.3.3. For crudely estimating the potential global-scale impact of our findings, we upscale the emission from boreal wetlands based on the average emissions measured in our study (in Sec. 3.2) and assuming it to be similar throughout summer (100 days). Using the wetland cover in the boreal environment (latitude above 50°N) of $5\times10^5$ km$^2$ (Junninen et al., 2022), we obtain total summer emissions of 0.4 Tg of isoprene, 0.12 Tg of each, MTs and SQTs, and 2 Gg of diterpenes. To compare with MEGAN emission estimates, we use the CLM4 land area of Arctic C3 Grass plant functional type ($5\times10^6$ km$^2$) and obtain total summer emissions of 4 Tg of isoprene, 1.2 Tg of each, MTs and SQTs, and 20 Gg of diterpenes during the summer of 2021. Meanwhile MEGAN estimates that boreal forests and shrubs (covering $24\times10^6$ km$^2$ of land area) emit 14 Tg of isoprene and 9 Tg of MTs (Guenther et al., 2012). The MEGAN emission estimate from boreal forests and shrubs is comparable to our Arctic C3 grass emission estimate despite the 5 times lower land cover.

**Minor comment:**

7. References should be douple checked for ACP style.

Reply: We have edited all the references now to ACP style.

**Reply to comments by RC2 (received 15 November 2022)**

**General comments:** The manuscript of "High emission rates and strong temperature response make boreal wetlands a large source of terpenes" by Vettikkat et al. measured terpenes flux by ecosystem-scale eddy covariance (EC) and Vocus-PTR from a wetland in southern Finland. The localized EF of isoprene, MTs and SQTs were obtained and compared with the MEGAN model. The research suggested that high-latitude ecosystems can be a significant source of terpenes to the atmosphere, and anthropogenic global warming could induce much higher BVOC emissions from wetlands. This manuscript is well written and recommend to be accepted after a minor revision.

Reply: We are grateful to the reviewer for their careful reading of the manuscript and comments and for recommending that the manuscript should be accepted after a minor revision.

 **Specific comments:**

Line 47-48, "Terpenes, except isoprene, have a huge variety of structures. They contain one or more double bonds making them highly reactive". Here "they" refers to "Terpenes" or "Terpenes, except isoprene", please rephrase clearly or combine into one sentence.

Reply: Replaced 'They' with 'Terpenes, except isoprene'.

Line 56-58, The ecological roles of BVOCs should be better phrased. For example, the related ecological interactions were missing here.

Reply: Added 'plant-pollinator interactions'.

Line 59, It may be easy to cause confusion if you include isoprene into terpenes. Here, temperature and light intensity are the main environmental drivers of isoprene emissions; as for MTs, the light-dependence may be more complex and highly dependent on species. The information on SQTs and DTs are even scarce. Their cases are largely different and may cause confusion when regarded as "terpene emissions".

Reply: We address the reviewer's concern by including isoprene when we talk about "terpenes" in this study (and point that out also explicitly in the text). We have specified this in the abstract, start of the introduction and conclusion. We have also edited the title as

"**High emission rates and strong temperature response make boreal wetlands a large source of isoprene and terpenes**".

However, specifically regarding Line 59, BVOC models like MEGAN generally use Temperature and PAR as the main driver for emissions of all terpenes, including isoprene. So, we have kept that sentence as it is.

Line 93-94, references should be added here.

Reply: We have added a reference (Helmig et al., 2004) as requested. We have also added '... they are emitted in low concentrations and ...' since

Line 100-104, could you provide some references regarding DTs here?

Reply: We have added a reference (Luo et al., 2021). We have also added the following text to extend the context of the discussion:

 Rose et al. (2018) indicated the importance of clustering of pure biogenic molecules to contribute during night-time in the boreal environment and recently Junninen et al. (2022) have found that high monoterpene emissions

and their subsequent oxidation will cause atmospheric clustering and new particle formation (NPF) in wetland environment at Siikaneva, Southern Finland.

Line 110-113, would you like to show some hypotheses here?

Reply: We expect the temperature response to be higher since the vegetation is not acclimatized to high temperatures. We add "... and hypothesize it to be higher than other ecosystems since boreal wetlands are not acclimatized to high temperatures." to the manuscript.

Line 273-276, here better to combine the sentences and use suitable conjunctions to flow better.

Reply: We rephrased the sentence as:

'However, Li et al. (2020) reported average mixing ratios of around 2 pptv in the French Landes Forest during summertime and Fischer et al. (2021) measured 0.1 pptv of DT above a boreal forest during spring.'

Line 300, what LoD stands for?

Reply: LoD stands for limit of detection. We have added the fullform.

Line 277, "3.2 Fluxes of terpenes from Siikaneva", this part I expect to see more links of your study with other studies, not only the descriptions of your study and previous studies in separate sentences.

Reply: We thank the reviewer for this comment, but it is difficult to consolidate all the results from different studies as the location, vegetation, and emissions are different. Thus, we use this section to simply reported the emissions along with the environmental conditions of previous studies to put our results into perspective.

Line 368, "3.3.1 Emission factors of terpenes", some symbols in the equations better be italic, the $R^2$ better be italic.

Reply: We have changed $C_L$, $C_T$ and $R_2$ and other symbols to italic.

Line 486, here you had some good discussion, maybe better appear in the above sections. Here it can be a concise conclusion section.

Reply: We would like this section to be a summary and conclusion and hence have repeated our key results again. We think the LAI determination, the drawbacks of current model and future outlook is better to be brought up here and would like to keep the section as it is.

**Other changes**

In addition to the changes requested by the reviewers we made some minor adjustments to further improve our manuscript. The changes are listed below and are clearly marked in the revised manuscript.

- We have added two additional authors to the author list: Tuukka Petäjä and Markku Kulmala from Institute for Atmospheric and Earth System Research, University of Helsinki, Finland.
- Moved GC results from Section 2.4 to Section 3.1
- Figure S5 is now S6 and viceversa.
- Added Seco et al. (2022) emissions to Section 3.2

- Edited Table 1 to add LAI to Eq.12 and MEGANv2.1 EF.
- Added corrected and MEGANv2.1 EF in units μg m$^{-2}$ hr$^{-1}$ and nmol m$^{-2}$ s$^{-1}$ for easier comparison.

**References**

Guenther, A. B., Jiang, X., Heald, C. L., Sakulyanontvittaya, T., Duhl, T., Emmons, L. K., and Wang, X.: The Model of Emissions of Gases and Aerosols from Nature version 2.1 (MEGAN2.1): an extended and updated framework for modeling biogenic emissions, Geoscientific Model Development, 5, 1471-1492, 10.5194/gmd-5-1471-2012, 2012.

Hellén, H., Schallhart, S., Praplan, A. P., Tykkä, T., Aurela, M., Lohila, A., and Hakola, H.: Sesquiterpenes dominate monoterpenes in northern wetland emissions, Atmospheric Chemistry and Physics, 20, 7021-7034, 2020.

Helmig, D., Bocquet, F., Pollmann, J., and Revermann, T.: Analytical techniques for sesquiterpene emission rate studies in vegetation enclosure experiments, Atmospheric Environment, 38, 557-572, https://doi.org/10.1016/j.atmosenv.2003.10.012, 2004.

Junninen, H., Ahonen, L., Bianchi, F., Quéléver, L., Schallhart, S., Dada, L., Manninen, H. E., Leino, K., Lampilahti, J., Buenrostro Mazon, S., Rantala, P., Räty, M., Kontkanen, J., Negri, S., Aliaga, D., Garmash, O., Alekseychik, P., Lipp, H., Tamme, K., Levula, J., Sipilä, M., Ehn, M., Worsnop, D., Zilitinkevich, S., Mammarella, I., Rinne, J., Vesala, T., Petäjä, T., Kerminen, V.-M., and Kulmala, M.: Terpene emissions from boreal wetlands can initiate stronger atmospheric new particle formation than boreal forests, Communications Earth & Environment, 3, 93, 10.1038/s43247-022-00406-9, 2022.

Luo, Y., Garmash, O., Li, H., Graeffe, F., Praplan, A. P., Liikanen, A., Zhang, Y., Meder, M., Peräkylä, O., Peñuelas, J., Yáñez-Serrano, A. M., and Ehn, M.: Oxidation product characterization from ozonolysis of the diterpene ent-kaurene, Atmos. Chem. Phys., 22, 5619-5637, 10.5194/acp-22-5619-2022, 2021.

Rose, C., Zha, Q., Dada, L., Yan, C., Lehtipalo, K., Junninen, H., Mazon, S. B., Jokinen, T., Sarnela, N., Sipilä, M., Petäjä, T., Kerminen, V. M., Bianchi, F., and Kulmala, M.: Observations of biogenic ion-induced cluster formation in the atmosphere, Sci Adv, 4, eaar5218, 10.1126/sciadv.aar5218, 2018.

Seco, R., Holst, T., Davie-Martin, C. L., Simin, T., Guenther, A., Pirk, N., Rinne, J., and Rinnan, R.: Strong isoprene emission response to temperature in tundra vegetation, 119, e2118014119, doi:10.1073/pnas.2118014119, 2022.